# Steel Corrosion Behavior of Reinforced Calcium Aluminate Cement-Mineral Additions Modified Mortar

**DOI:** 10.3390/ma14144053

**Published:** 2021-07-20

**Authors:** Zhongping Wang, Yuting Chen, Zheyu Zhu, Xiang Peng, Kai Wu, Linglin Xu

**Affiliations:** 1School of Materials Science and Engineering, Tongji University, Shanghai 201804, China; wangzpk@tongji.edu.cn (Z.W.); 1930649@tongji.edu.cn (Y.C.); 17712937215@163.com (Z.Z.); 1830658@tongji.edu.cn (X.P.); wukai@tongji.edu.cn (K.W.); 2Key Laboratory of Advanced Civil Engineering Materials, Tongji University, Ministry of Education, Shanghai 201804, China

**Keywords:** mineral additions modified calcium aluminate cement, steel reinforcement, chloride corrosion, electrochemical impendence

## Abstract

Mineral additions can eliminate the conversion in calcium aluminate hydrates and thus inhibit the future strength retraction of calcium aluminate cement (CAC). However, the impacts of these additions on the protection capacity of CAC concrete in relation to the corrosion of embedded steel reinforcement remains unclear. This paper focused on the corrosion behavior of steel reinforcement in slag, limestone powder, or calcium nitrate-modified CAC mortars via XRD and electrochemical methods (corrosion potential, electrochemical impedance, and linear polarization evaluation). The results indicate that strätlingite (C_2_ASH_8_), which is formed in slag-modified CAC, has poor chloride-binding ability, leading to decline in corrosion resistance of the steel reinforcement. The electrochemical parameters of specimens immersed in NaCl solution suddenly drop at 14 days, which is 28 days earlier than that of the references. In contrast, the Ca_2_[Al(OH)_6_]_2_0.5CO_3_OH·H_2_O (CaAl·CO_3_^2−^-LDH) and 3CaO·Al_2_O_3_·Ca(NO_3_)_2_·12H_2_O (NO_3_-AFm) in limestone powder and calcium nitrate-modified CAC mortar show great chloride-binding ability, thereby improving the corrosion resistance of the steel reinforcement. The electrochemical parameters of specimens modified with calcium nitrate maintain a slow decreasing trend within 90 days.

## 1. Introduction

In marine engineering constructions, the foundation often comprises of reinforced concrete structures. However, damages caused by seawater corrosion have led to huge economic losses and waste of resources [1]. Chloride corrosion is the main cause for this, as it destroys the passive film on the surface of the steel reinforcement. The large volumes of rust lead to crack in the concrete under expansion stress, thus accelerating chloride corrosion rates [2,3,4]. Studies have shown that reducing free chloride content is the most effective way to protect steel reinforcements in concrete; thus, chloride-binding ability is of great significance [5,6,7].

CAC has been known for excellent chemical resistance since its inception and was a breakthrough in marine constructions [8]. Compared with Portland cement, CAC is superior in chloride-binding capacity, owing to its higher aluminate content [9]. Cl^-^ directly reacts with both unhydrated and hydrated phases in CAC, and then forms Friedel’s salt, leading to a decrease in free chlorides [10]. Friedel’s salt refines pore structures and enhances the impermeability of the structure, thereby preventing further intrusion of Cl^−^. However, the hydration of CAC is highly sensitive to temperature. High temperatures promote phase conversion from unstable hexagonal calcium aluminate hydrates (C_2_AH_8_ and CAH_10_) to a stable cubic one (C_3_AH_6_) in CAC, resulting in strength retraction. Therefore, since the 1970s, CAC has been banned from being used in structural constructions [11,12].

Many research works have focused on solving the problem of pore structure deterioration and strength retraction in CAC by adding slag [13,14], limestone powder [15,16,17,18], calcium nitrate [19], and other mineral or salt additives. These additives can change CAC hydrates, thus effectively avoiding its later phase transition. Slag can promote the formation of straetlingite (C_2_ASH_8_) in CAC, and enhance high temperature resistance and subsequent strength development [13,14]. Limestone powder can react with CAC to produce carbonaluminate hydrates with high strength and volume stability [15,16,17]. The strength of the mortar is the highest when the mixing content of limestone powder is 3% [18]. When mixing with calcium nitrate, lower Gibbs free energy nitrate-ettringite (NO_3_-AFm and NO_3_-AFt) is formed instead of unstable phases, thereby greatly improving the compressive strength of CAC [19].

Although several studies have investigated the impact of mineral additions on the strength development of CAC, seldom have focused on their effects on its corrosion resistance. Therefore, this paper aims to clarify the chloride corrosion of steel reinforcements in CAC modified by mineral additions. The mechanism of these mineral additions on the corrosion resistance of CAC was explored by measuring phase compositions and electrochemical methods from the aspects of corrosion potential, electrochemical impedance spectroscopy, and linear polarization. It provides theoretical support for CAC with excellent seawater corrosion resistance to better serve in marine engineering constructions.

## 2. Materials and Methods

### 2.1. Raw Materials

The CAC used in this study was supplied by Imerys Aluminates (Tianjin, China) with grade CA50. Its chemical composition measured by XRF is given in Table 1. The main mineral composition of CAC tested by QXRD is summarized in Table 2. Q235 ordinary round steel bar was used as the embedded steel, and its tensile strength was 412 MPa. The chemical composition of the steel bar is shown in Table 3. Sand and deionized water were also employed. Chemically pure NaCl reagent was utilized as the chloride source. S95 slag (chemical composition is given in Table 4), limestone powder, and chemically pure calcium nitrate (Ca(NO_3_)_2_·4H_2_O) were used as mineral additions.

### 2.2. Specimen Preparation

To accelerate corrosion, the water-to-cement ratio was fixed at 0.55, according to the regulations in GB/T 17671-1999 (ISO679:1989) “Cement Mortar Strength Test Method”, and the cement-to-sand ratio was 1:3. The CAC pastes were modified with slag, limestone powder, and calcium nitrate, according to the mix proportion in Table 5. One specimen was prepared for each set of samples, and tested at a certain age. Before molding, the surface of the steel reinforcement was polished until bright silvery white metallic luster was emitted, and then the steel bars were ultrasonically cleaned with absolute ethanol and dried with a cold air blower. One end of the steel bars (with diameter of 10 mm and height of 50 mm) were welded with copper wires and then the upper and lower ends of the steel bars were coated with epoxy. The cylindrical side of each steel bar was the working surface with an area of 15.71 cm^2^. Prior to being embedded in the concrete, the steel bars were immersed in alkaline solution (0.6 mol/L KOH + 0.2 mol/L NaOH) to promote passivation. The reinforced cylindrical specimens (with diameter of 70 mm and height of 60 mm) were then prepared to study the corrosion behavior of steel bars embedded in CAC mortars [20].

All specimens were cured at 40 ± 2 °C and relative humidity 65 ± 2% for 24 h, followed by curing in water for 28 days. Then the two ends of the specimens were coated in epoxy twice to ensure that only the cylindrical sides of the specimens are directly in contact with the chloride salt solution in the corrosion test. Figure 1 and Table 6 show the XRD pattern and pore parameters of the specimens.

### 2.3. Pore Paremeters

Twenty-eight-day specimens were broken into spherical pieces with a diameter of about 8 mm after vacuum drying. The pore parameters of the samples were tested by the mercury intrusion method (MIP) using American Quantachrome AUTOSCAN-60.

### 2.4. Chloride Corrosion

The above-mentioned specimens, after being pre-cured for 28 days under 40 °C, were kept immersed in chloride salt solution at a concentration of 10% (about 3 times the concentration of chlorides in seawater) at 20 ± 2 °C. Three specimens were chosen as a set, and every month, the solutions were changed. The volume ratio of each set of specimens to the solution was 1:20. Electrochemical tests were regularly carried out on the immersed specimens.

### 2.5. Electrochemical Test

The classic three-electrode system was utilized in the electrochemical test. Each reinforced CAC specimen acted as a working electrode. A saturated calomel electrode and a platinum electrode were selected as the reference electrode and the counter electrode, respectively. The three electrodes were connected with a CHI 660E electrochemical workstation and a computer to form a testing system. All test results were derived from the average of three specimens in each set.

After connecting the circuit, corrosion potential was measured by open circuit potential (OCP) test procedure in the electrochemical workstation. The corrosion potential of the specimen was the potential difference between the specimen and the saturated calomel electrode. If the OCP did not exceed ±2 mV within 5 min, the corrosion system was considered to be stable, and the final value was identified as the corrosion potential.

The electrochemical impedance spectroscopy (EIS) test was performed after the corrosion potential testing. It was obtained under a frequency range from 10^5^ Hz to 10^−2^ Hz. In order to quantitatively analyze the microstructure of the CAC mortar and the electrochemical characteristics of the steel reinforcement during the corrosion process, the ZSimpWin software program (https://echem-software-zsimpwin.software.informer.com/ accessed on 15 July 2021) was used to provide the equivalent circuit model for analyzing the EIS data. After comprehensively comparing the fitting effects of various classical equivalent circuit models [21,22,23,24,25], the model *R*_s_(*Q*_c_*R*_c_)(*Q*_dl_(*R*_ct_*W*)) was used to fit the impedance spectroscopy of the reinforced CAC mortar for chloride ion attack.

The corrosion current density can quantitatively characterize the corrosion degree of the reinforcement, which is calculated according to Equation (1) [26,27]:(1)$$i_{\mathrm{corr}}=B/R_{p}$$
where, *B* is a constant potential of 10 mV. To measure *R*_p_, the linear polarization curves were obtained using the linear sweep voltammetry (LSV) method. The polarization results were then approximated to a straight line, and the slope of the straight line is the reciprocal of *R*_p_.

### 2.6. Phase Composition

The compositions of the specimens were tested by XRD and TG-DSC. XRD data were obtained using a Rigaku-D/max 2550VB3 + X-ray diffractometer with a graphite-monochromatized Cu Kα radiation (*λ* = 1.541 Å) generated at 40 kV and 100 mA, to qualitatively identify the main crystalline phases in the hardened CAC pastes. Continuous scanning was performed with 2*θ* ranging from 5° to 75° and scanning speed of 2°∙min^−1^.

## 3. Results and Discussion

### 3.1. Corrosion Potential of Steel Reinforcement

Figure 2 shows the relationship between the corrosion potential of different CAC mortar-rebar electrodes immersed in NaCl solution over time. All specimens showed a significant decline in corrosion potential after one day, owing to the water-saturation process of the pastes. For the control, the corrosion potential was kept stable at around 250 mV within 14 days. A substantial decrease in corrosion potential was observed after 28 days, dropping to 500 mV at 56 days, which indicates that the passive film on the surface of the steel reinforcement in the specimen was destroyed at 28 days, and the steel reinforcement started to get corroded. The stabilized corrosion potential shows a slower corrosion rate.

The corrosion potential of the slag (SL-30) and limestone (LS-30) modified CAC decrease continuously as the immersion age increases, only achieving approximately 200 mV after 28 days, indicating that the passive film was corroded from the beginning and severely damaged by 28 days. With increase in immersion age, the corrosion potential of the SL-30 specimen declines slowly with a longer period, which means that the corrosion is slow. However, the corrosion potential of the LS-30 specimen drops significantly after 42 days, with only about 700 mV at 90 days, indicating an increasing corrosion at later stages. In contrast, although the corrosion potential of the calcium nitrate-modified CAC specimen (CN-10) is the lowest on day 1, it is found to be stable at about 400 mV in the long term. This finding reveals that the passive film of the steel reinforcement is stable, with no corrosion inside.

Comparing the corrosion potential change between four specimens, it was found that the addition of 30% slag and limestone does not improve the corrosion resistance of CAC, but accelerates the corrosion rate of the steel reinforcement; the addition of 10% calcium nitrate exhibits a promotion effect, owing to its optimization on the pore structure, which significantly reduces the average pore diameter.

### 3.2. Electrochemical Impedance Spectroscopy

The EIS spectra of the reinforced CAC mortar and different reinforced CAC-mineral additions-modified mortar electrodes are shown in Figure 3, Figure 4 and Figure 5. The impedance arcs in the low-frequency zone of the EIS curve reflect the electrochemical properties of passive film. The impedance arc radius (Nyquist curve), the impedance module, and the phase angle (Bode curve) in the low frequency zone are inversely proportional to the corrosion rate of the steel reinforcement [28].

As shown in Figure 3a, when the control specimens are immersed for 14 days, the impedance arcs in the low-frequency zone are similar to a straight line with a slope greater than 1. This infers that the passive film of the steel reinforcement is stable at this age. After being immersed for 28 days, the impedance arc radius in the low-frequency zone is found to have decreased sharply, indicating that the passive film is destroyed. The charge transfer resistance dropped suddenly, and the steel bar was found to be corroded. With increase in immersion time, the impedance arc radius in the low-frequency zone keeps decreasing. However, it increases at 90 days, which means that the corrosion rate decreases at a later stage. The results of the Bode curve in Figure 3b are consistent with the Nyquist curve. The impedance modulus and phase angle of the specimen in the low frequency zone change slowly within 14 days of immersion, and the impedance modulus declines rapidly to 30 kΩ·cm^2^ at 28 days, accompanied by significant decline in phase angle, indicating that the passive film starts to get damaged. The impedance modulus and phase angle decrease with immersion age, until a revival at 90 days, owing to the reduction of Friedel’s salt on chloride corrosion by optimizing pore structure.

For the SL-30 specimens (Figure 4a), the impedance arc radius in the low-frequency zone begin to decrease after being immersed for seven days, which reveals that the passive film is unstable and dissolves gradually. The impedance arc radius reduces after 28 days of immersion, and the surface of the steel reinforcement is active, with continuous corrosion. Correspondingly, the impedance modulus and phase angle in the Bode curve (Figure 4b) decreases with age, which further confirms that the corrosion of the steel reinforcement in slag-modified CAC occurs earlier.

As shown in Figure 5a, the impedance arc radius of the LS-30 specimens in the low frequency zone of the Nyquist curve are stable within seven days of immersion, displaying straight segments with a slope over 1, which indicates that the passive film is in a relatively stable state. However, the impedance arc suddenly decreases at 14 days, indicating that the passivation film starts to get destroyed. Within 56 days, the smaller trend of the impedance arc slows down; however, the decrease is obvious again at 90 days, which is consistent with the corrosion potential results (Figure 2). This finding indicates that the passivation film is further corroded in a later stage, with deepening corrosion. The phase angle and impedance modulus in the Bode curve (Figure 5b) have a consistent trend as the impedance arc, which confirms that 30% limestone accelerates the chloride corrosion of CAC instead of improving its resistance.

For the CN-10 specimens (Figure 6a), the impedance arc radius in the low frequency zone of the Nyquist curve was found to be stable within 28 immersion days, indicating that the passive film is stable. After immersion for 90 days, the impedance arc gradually reduces, but the end of the impedance arc is still straight, indicating that the passive film is not dissolved and penetrated, so the steel bar is still in a relatively safe state. As shown in Figure 6b, the impedance modulus of the steel bar electrode slowly decreases in 90 days, with the lowest impedance modulus around 50 kΩ∙cm^2^. In addition, the phase angle remains at around 50°, unchanged with age. Both the Nyquist curve and the Bode curve illustrate that calcium nitrate promotes the chloride corrosion resistance of CAC mortar.

### 3.3. Linear Polarization

For further quantitative analysis, the linear polarization fitting results are summarized in Table 7. It is noted that before the corrosion (0 d), the polarization resistivity *R_p_* of each set was around 4~11 kΩ·cm^2^, and the linear correlation coefficient *R*^2^ ranged from 0.8 to 0.9. The linear polarization resistance *R_p_* of each steel bar electrode gradually decreases with age, and the corresponding linear correlation coefficient *R^2^* also increases gradually. With immersion, the changing trend of *R_p_* and *R*^2^ in four sets varies from each other.

For the CAC specimens, the *R_p_* decreases gently within 14 days, with no obvious increase in corresponding *R*^2^, indicating that the passivation film on the surface of the reinforcement is not damaged by chloride ion corrosion. However, the *R_p_* decreases by more than 50% at 28 days and *R*^2^ goes up to 0.89, showing great corrosion of the passive film. The corrosion gradually slows down by 90 days, owing to the optimization of Friedel’s salt on the pore structure. This result is in agreement with that of electrochemical impedance spectroscopy.

The SL-30 specimen and the LS-30 specimen have similar changing trends of *R_p_* and *R*^2^. In the first seven days, *R_p_* of both specimens decrease by more than 50%, with *R*^2^ obviously increasing, indicating that the passive film on the surface of the reinforcement is corroded and dissolved in the beginning. As immersion age increased to 90 days, the polarization resistance of the SL-30 specimen and the LS-30 specimen continuously decline to merely 0.52 KΩ∙cm^2^ and 1.46 KΩ∙cm^2^, and their linear correlation coefficient increase to 0.95 and 0.93, respectively. It can be concluded that neither 30% slag nor limestone powder is conducive to improving the corrosion resistance of CAC.

In contrast, although the initial *R*^2^ of the CN-10 specimen is higher than that of the other three specimens, it is stable within 90 days. Polarization resistance *R_p_* also shows a slow downward trend with age. After immersion for 42 days, *R_p_* is still over half of that at one day. It is superior to the other groups, which proves that the calcium nitrate modification helps protect the steel reinforcement.

The EIS spectra of the reinforced CAC mortar and different reinforced CAC-mineral additions modified mortar electrodes are shown in Figure 3, Figure 4 and Figure 5. The impedance arcs in the low-frequency zone of the EIS curve reflect the electrochemical properties of passive film. The impedance arc radius (Nyquist curve), the impedance module, and the phase angle (Bode curve) in the low frequency zone are inversely proportional to the corrosion rate of the steel reinforcement [28].

### 3.4. Phase Composition

Figure 7 shows the XRD patterns of the CAC, SL-30, LS-30, and CN-10 specimens after being immersed in 100 g/L NaCl solution for 90 days. Massive amounts of calcium carbonate were found in all samples, showing that all specimens were carbonized. Compared to the initial composition (Figure 1), massive forms of Friedel’s salt were found in the CAC specimen, while the C_3_AH_6_ was reduced correspondingly, illustrating that some of the C_3_AH_6_ is chemically combined with the chlorides.

C_2_ASH_8_ is still visible in the SL-30 specimen after immersion, due to its difficulty of binding with chlorides. In addition, no Friedel’s salt is generated after corrosion, while substantial C_3_AH_6_ is present. This indicates that C_2_ASH_8_ inhibits the chloride binding of C_3_AH_6_, explaining the poor chloride resistance of the slag-modified CAC specimens.

A certain amount of Friedel’s salt is generated in the LS-30 specimen modified with limestone, and CaAl·CO_3_^2−^-LDH almost disappears. It can be inferred that CaAl·CO_3_^2−^-LDH has strong chloride-binding ability. However, excessive limestone reduces the chloride resistance of CAC. Thereby, the steel reinforcement begins to corrode in early stages.

In the CN-10 specimen, the initial hydration products, NO_3_-AFm and C_3_AH_6_, disappear after chloride corrosion, and substantial amount of Friedel’s salt is generated instead. Both AFm phases and Friedel’s salt are members of the LDH supergroup and their lyotropic order for the interlayer anion exchange affinity is CO_3_^2−^ > F^−^ > Cl^−^ > Br^−^ > ClO^4−^ > NO^3−^ > I^−^), which leads to a higher chloride binding ability of the NO_3_-AFm phases, compared to the carbonate-containing ones [29,30]. This shows that the formation of NO_3_-AFm can compensate for the adverse effects on the pore structure caused by the phase transition, thus improving the chloride corrosion resistance of CAC. These results are highly consistent with the results of electrochemical tests, confirming that calcium nitrate benefits CAC in corrosion resistance.

## 4. Conclusions

In this work, the electrochemical corrosion behavior of steel reinforcement in CAC mortar modified by mineral additions was evaluated by means of XRD and electrochemical methods, which include corrosion potential, electrochemical impedance, and linear polarization evaluation. Based on the experimental results, the main conclusions obtained are as follows:
The electrochemical parameters of calcium nitrate-modified CAC specimens show a slow downward trend within 90 days, earlier than the unmodified CAC specimen. This indicates that calcium nitrate can improve the corrosion resistance of steel reinforcements in CAC mortar. However, slag and limestone powder show adverse effects on the corrosion resistance of steel bars in CAC mortar. The relevant electrochemical parameters drop suddenly at 14 days, leading to early corrosion.The NO_3_-AFm in the calcium nitrate-modified CAC specimen was found to have great chloride-binding ability by generating substantial amount of Friedel’s salt, which improves the chloride resistance of the CAC paste. The CO_3_-AFm in the limestone powder-modified CAC specimen also can combine with chloride to form Friedel’s salt. However, the chloride-binding ability of C_2_ASH_8_ in the slag-modified CAC specimen is poor, and thus not conducive to corrosion resistance of the CAC paste.

## Figures and Tables

**Figure 1 materials-14-04053-f001:**
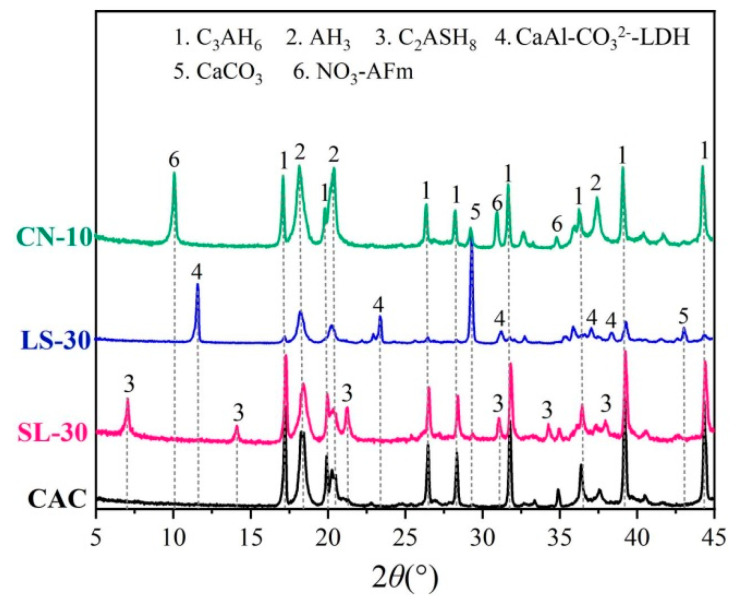
XRD pattern of CAC mortars (modified by mineral additions) cured at 40 °C for 28 days.

**Figure 2 materials-14-04053-f002:**
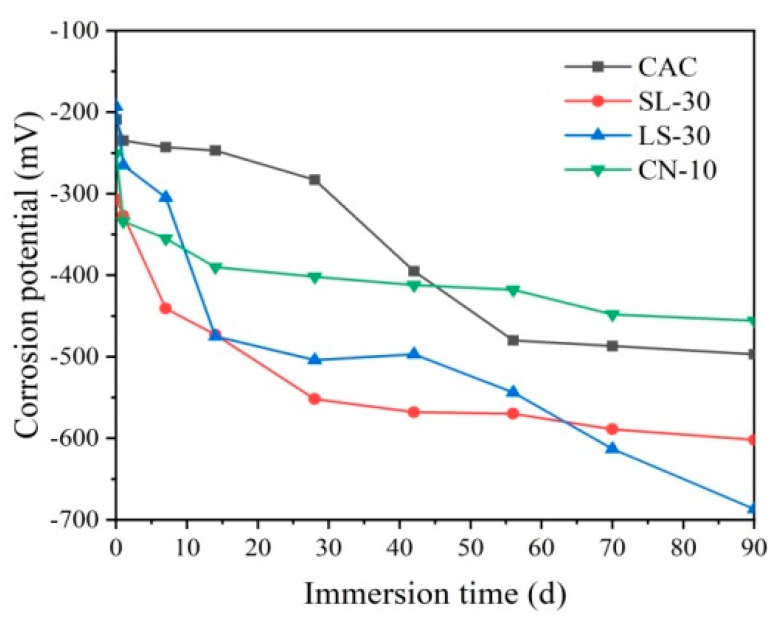
Corrosion potential of different reinforced CAC mortar electrodes.

**Figure 3 materials-14-04053-f003:**
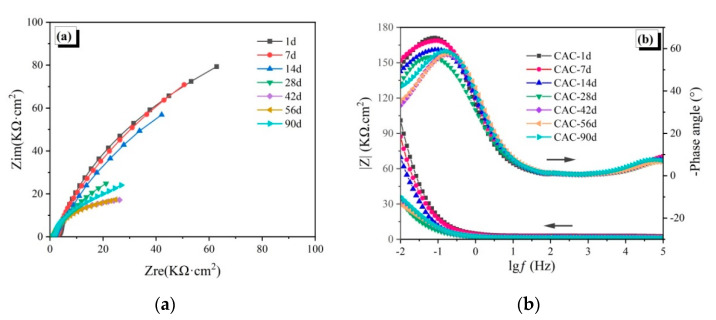
Electrochemical impedance spectroscopy of the CAC specimens: (**a**) Nyquist curves, (**b**) Bode curves.

**Figure 4 materials-14-04053-f004:**
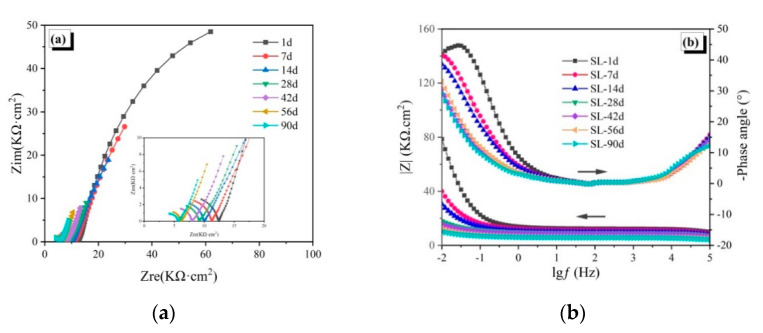
Electrochemical impedance spectroscopy of the SL-30 specimens: (**a**) Nyquist curves, (**b**) Bode curves.

**Figure 5 materials-14-04053-f005:**
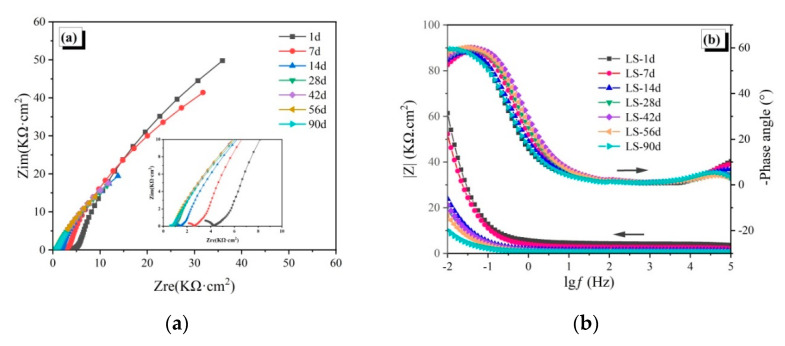
Electrochemical impedance spectroscopy of the LS-30 specimens: (**a**) Nyquist curves, (**b**) Bode curves.

**Figure 6 materials-14-04053-f006:**
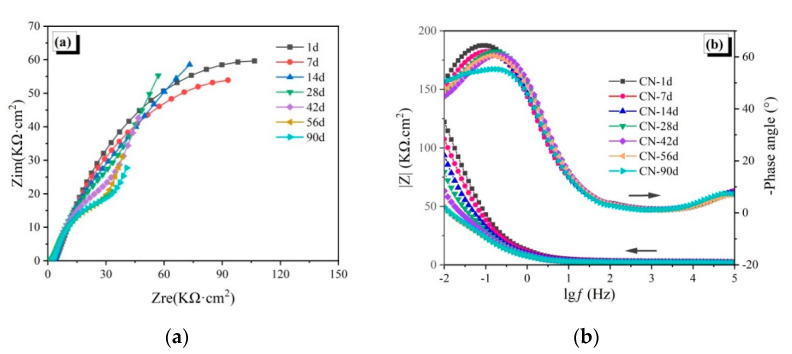
Electrochemical impedance spectroscopy of the CN-10 specimens: (**a**) Nyquist curves, (**b**) Bode curves.

**Figure 7 materials-14-04053-f007:**
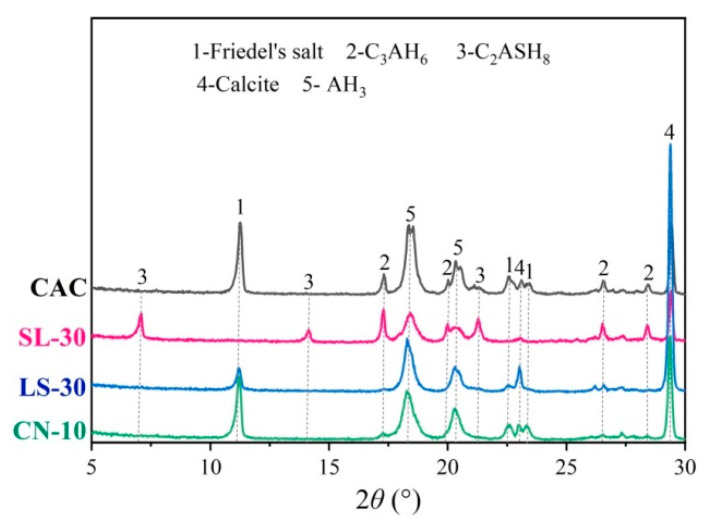
XRD pattern of modified CAC pastes immersed in 100 g/L NaCl solution for 90 days.

**Table 1 materials-14-04053-t001:** Chemical composition of CAC (wt %).

MgO	Al_2_O_3_	SiO_2_	P_2_O_5_	SO_3_	Cl	K_2_O	CaO	TiO_2_	MnO	Fe_2_O_3_
0.46	48.45	7.33	0.15	0.46	0.01	0.45	37.87	2.52	0.04	1.90

**Table 2 materials-14-04053-t002:** Mineralogical composition of CAC (wt %).

CA	CA_2_	C_2_AS	CT
51.12	4.31	35.15	4.86

**Table 3 materials-14-04053-t003:** Chemical composition of the steel bar (wt %).

C	Si	Mn	P	S	Ni	Cr	Cu	Fe
0.140	0.180	0.330	0.0170	0.004	0.010	0.010	0.010	99.0

C

**Table 4 materials-14-04053-t004:** Chemical composition of S95 slag (wt %).

SiO_2_	CaO	Al_2_O_3_	Fe_2_O_3_	MgO	K_2_O	SO_3_	Na_2_O
32.22	39.03	17.01	0.43	7.67	0.27	2.08	0.42

**Table 5 materials-14-04053-t005:** Mix proportion of the pastes used for the mortar-reinforcement specimen (wt.%).

	CAC	Slag	Limestone	Calcium Nitrate
CAC	100	0	0	0
SL-30	70	30	0	0
LS-30	70	0	30	0
CN-10	90	0	0	10

**Table 6 materials-14-04053-t006:** Pore parameter of CAC mortars (modified by mineral additions) cured at 40 °C for 28 d.

Samples	Pore Volume Fraction/%			Porosity/%	Average Pore Diameter/nm
	<100 nm	100~1000 nm	>1000 nm		
CAC	7.40	23.86	68.74	11.82	400.20
SL-30	16.65	26.32	57.03	13.11	79.00
LS-30	28.32	28.97	42.71	16.63	40.90
CN-10	41.99	45.78	12.23	15.02	30.70

**Table 7 materials-14-04053-t007:** Mix proportion of the pastes used for the mortar-reinforcement specimens (wt.%).

Samples	Fitting Terms	Immersion Time						
		1 d	7 d	14 d	28 d	42 d	56 d	90 d
CAC	*R_p_* (KΩ·cm^2^)	7.81	6.55	4.93	2.43	1.87	1.76	1.86
	*R* ^2^	0.84	0.83	0.86	0.89	0.93	0.94	0.94
SL-30	*R_p_* (KΩ·cm^2^)	4.15	1.98	1.49	0.93	0.80	0.59	0.52
	*R* ^2^	0.79	0.81	0.85	0.93	0.93	0.94	0.95
LS-30	*R_p_* (KΩ·cm^2^)	10.81	4.29	2.32	2.24	1.72	1.60	1.46
	*R* ^2^	0.84	0.90	0.92	0.92	0.92	0.92	0.93
CN-10	*R_p_* (KΩ·cm^2^)	7.05	6.15	5.50	4.88	3.68	2.87	2.50
	*R* ^2^	0.90	0.92	0.89	0.89	0.90	0.93	0.91

## Data Availability

Data available on request due to restrictions e.g., privacy or ethical.

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
