# Peer review of "Steel Corrosion Behavior of Reinforced Calcium Aluminate Cement-Mineral Additions Modified Mortar"

_materials, 2021, doi:10.3390/ma14144053_

Round 1
Reviewer 1 Report
Dear Authors,
The paper Steel corrosion behavior of reinforced calcium aluminate cement-mineral additions modified mortar by Zhongping Wang, Yuting Chen, Zheyu Zhu, Xiang Peng, Kai Wu, Linglin Xu is well suited for journal Materials. The authors of this article analyzed the results of the present studies on seawater corrosion resistance of steel reinforcement in a variety of CAC mortars.
The paper is interesting and scientifically valuable. The paper contains parts in good, typical order: introduction, materials and methods, results, conclusions.
General comments:
- How many sets (row 117-118) of specimens were made, how many were used to establish individual values? How the variability of the results was analyzed?
- Was the axial position of the bars in the speciments ensured? It is worth giving a comparison of the cover thickness in speciments to the standard requirements (in the context of various standards) for the conditions simulated in the tests.
- Did the authors consider how changing the cover thickness would affect the research results, mainly in the context of the idea that with certain cover thicknesses and tight concrete, the influence of penetration of corrosive factors can be almost extinguished?
- The assumed w/c value was given, but no reference was made to what values recommended by standards. In the opinion of the reviewer, the issue of "unsealing the structure" with pores and their size should be taken into account. With regard to the cover thickness of the bars, it should be considered whether the conditions are too favorable for penetration. Of course, it was necessary to speed up the experiment.
Detailed comments:
Not noticed, the article was written carefully.
The article was written enough well in English, is understandable for a reviewer, a person who does not speak English as a mother tongue.
Author Response
- How many sets (row 117-118) of specimens were made, how many were used to establish individual values? How the variability of the results was analyzed?
- One specimen is prepared for each set of samples, and being test at certain age. So no variability of results needs to be analyzed.
- Was the axial position of the bars in the specimens ensured? It is worth giving a comparison of the cover thickness in specimens to the standard requirements (in the context of various standards) for the conditions simulated in the tests.
- The axial positions of bars in the specimens were ensured. And the cover thickness of specimens is 30 mm according to other reinforced corrosion researches such as “Study on corrosion durability with electrochemical tests of GGBS/Portland blends activated by chlorides.”
- Did the authors consider how changing the cover thickness would affect the research results, mainly in the context of the idea that with certain cover thicknesses and tight concrete, the influence of penetration of corrosive factors can be almost extinguished?
- The increasing porosity of concrete enhances the permeability of ions, thereby increasing the corrosion rate of chlorides. The denser the concrete, the less the corrosion of the steel bar will be. But the corrosion will also deepen over time, so this problem cannot be ignored. Therefore, this paper controls the porosity by fixing the water-cement ratio, and focuses on the corrosion of steel bars in concrete with the same porosity.
- The assumed w/c value was given, but no reference was made to what values recommended by standards. In the opinion of the reviewer, the issue of "unsealing the structure" with pores and their size should be taken into account. With regard to the cover thickness of the bars, it should be considered whether the conditions are too favorable for penetration. Of course, it was necessary to speed up the experiment.
- The molding of the mortar test piece is carried out in accordance with the regulations in GB/T 17671-1999 (ISO679:1989) "Cement Mortar Strength Test Method". The W/C of CA50 need to be higher than 0.44, and the researches in other papers use W/C range from 0.5 to 0.7, so in our paper, we choose 0.55 in the middle.

Reviewer 2 Report
The ms titled “Steel corrosion behavior of reinforced calcium aluminate cement-mineral additions modified mortar” aims to show an alternating way of using CAC that was banned since the 1970s and used to improve the corrosion resistance of steel reinforcement. I think the paper is straightforward. However, the authors mentioned under the Materials and Methods section TG-DSC. TG-DSC is missing from results and discussion. Can authors provide results and discussion for this?
Line 17 – what is the meaning C2ASH8?
Line 20 - CO3-AFm and NO3-AFm – write it indicate
Line 56-57 - And the strength of the mortar is the highest when limestone powder content is 3% [18] – please check the line, the statement is not complete.
Author Response
- However, the authors mentioned under the Materials and Methods section TG-DSC. TG-DSC is missing from results and discussion. Can authors provide results and discussion for this?
- Thank you for reminding. I am sorry that TG-DSC test has not been applied in the research of this paper, so I have deleted the method from the “Materials and Methods” section.
- Line 17 – what is the meaning C2ASH8?
- C2ASH8 is strätlingite, and the name has been added.
- Line 20 - CO3-AFm and NO3-AFm – write it indicate
- CO3-AFm and NO3-AFm has been write indicate as 3CaO·Al2O3CaCO3·12H2O and 3CaO·Al2O3·Ca(NO3)2·12H2O.
- Line 56-57 - And the strength of the mortar is the highest when limestone powder content is 3% [18] – please check the line, the statement is not complete.
- The statement is revised as “And the strength of the mortar is the highest when the mixing content of limestone powder is 3%

Reviewer 3 Report
The paper deals with the development of the calcium aluminate cement using commercially available additives like limestone, calcium nitrate and waste slag. The authors applied several characterization techniques (XRD, TG-DSC, textural and electrochemical analysis), some part of the work is maybe of interest, but, several confusing mistakes make difficult the interpretation, the manuscript in its present form requires major revision. The corrected version of the manuscript can only be accepted, if all remarks are answered carefully.
- The reviewer ask the authors to avoid the use of the abbreviation without accurate description especially in the abstract section.
- From the materials and methods section, the technical data for the determination of pore parameters are missing.
- In the Table 4., the explanation of the PC denotation is missing and it is not clear why it is important to sign one by one the absence of it in all cases. For CN-10 row, the 110 wt% composition is faulty, probably, the 90 wt% for CAC would be the right form.
- The chemical description of the used additives is insufficient; hydration sate of the calcium nitrate, and the chemical composition of the slag are required.
- For the assignation of the reflections on the X-ray diffraction patterns (Figure 1 and 7), the reviewer would like to see the used references. Please note that, the formation of the AFm phases with hydroxide−carbonate interlayer binary anions are also extremely likely in this environment. The reflections at around 12° and the doubled ones around 22° − 24° 2 theta indicate rather the formation of the hemicarbonated than the monocarbonated AFm for the sample named LS-30 doi:10.1016/j.ultsonch.2016.01.026.
- It would be advisable to introduce the term of layered double hydroxide (LDH), since the AFm phases and Friedel’s salt prepared are members of the LDH supergroup and their long ago determined lyotropic order for the interlayer anion exchange affinity (doi.org/10.1016/j.ultsonch.2017.08.041) elucidate better the higher chloride binding ability of the NO3-AFm phases compared to the carbonate containing ones have (carbonate anions are at the beginning of the order, while the nitrate is in the back). Please consider this.
- Finally, it is not clear how the TG-DSC measurements were utilized in the manuscript.
Author Response
# Reviewer 3
- The reviewer ask the authors to avoid the use of the abbreviation without accurate description especially in the abstract section.
- The abstract has checked and all abbreviations have been revised.
- From the materials and methods section, the technical data for the determination of pore parameters are missing.
- The pore parameters test methods was added. “The specimens of 28 days were broken into spherical pieces with a diameter of about 8 mm and after vacuum drying. And the parameters of the samples were test by the mercury intrusion method (MIP), using Quantachrome AUTOSCAN-60.”
- In the Table 4., the explanation of the PC denotation is missing and it is not clear why it is important to sign one by one the absence of it in all cases. For CN-10 row, the 110 wt% composition is faulty, probably, the 90 wt% for CAC would be the right form.
- The PC denotation is not necessary and it has been removed from table 4.
- The CN-10 row, the CAC content has been revised as 90 wt%.
- The chemical description of the used additives is insufficient; hydration sate of the calcium nitrate, and the chemical composition of the slag are required.
- The calcium nitrate is Ca(NO3)24H2O.
- The Chemical composition of S95 slag is given in Table 4.
Table 4. Chemical composition of S95 slag (wt.%).
SiO2 |
CaO |
Al2O3 |
Fe2O3 |
MgO |
K2O |
SO3 |
Na2O |
32.22 |
39.03 |
17.01 |
0.43 |
7.67 |
0.27 |
2.08 |
0.42 |
- For the assignation of the reflections on the X-ray diffraction patterns (Figure 1 and 7), the reviewer would like to see the used references. Please note that, the formation of the AFm phases with hydroxide−carbonate interlayer binary anions are also extremely likely in this environment. The reflections at around 12° and the doubled ones around 22° − 24° 2 theta indicate rather the formation of the hemicarbonated than the monocarbonated AFm for the sample named LS-30 doi:10.1016/j.ultsonch.2016.01.026.
- Through the futher analysis of the reflection according to reference, the monocarbonated AFm has been revised to hemicarbonated AFm in the XRD.
- It would be advisable to introduce the term of layered double hydroxide (LDH), since the AFm phases and Friedel’s salt prepared are members of the LDH supergroup and their long ago determined lyotropic order for the interlayer anion exchange affinity (doi.org/10.1016/j.ultsonch.2017.08.041) elucidate better the higher chloride binding ability of the NO3-AFm phases compared to the carbonate containing ones have (carbonate anions are at the beginning of the order, while the nitrate is in the back). Please consider this.
- The lyotropic order for the interlayer anion exchange affinity of LHD in the references has been applied to explain as “Both AFm phases and Friedel’s salt are members of the LDH supergroup and their lyotropic order for the interlayer anion exchange affinity is CO32− > F− > Cl− > Br− > ClO4− > NO3− > I−), which leads to the higher chloride binding ability of the NO3-AFm phases compared to the carbonate containing ones.”
- Finally, it is not clear how the TG-DSC measurements were utilized in the manuscript.
- Thank you for reminding. I am sorry that TG-DSC test has not been applied in the research of this paper, so I have deleted the method from the “Materials and Methods” section.

Round 2
Reviewer 3 Report
The authors answered for all comments, the manuscript has been developed significantly. The corrected version of the manuscript in its present form requires only minor revision.
Now, the manuscript contains three Figure 4, a renumbering is required!
Author Response
Thankyou for reminding! All table number has been checked and renanmed in the manuscript.
